# News source bias and sentiment on social media

**Brian Knutson** **\*, Tiffany W. Hsu, Michael Ko, Jeanne L. Tsai**

Department of Psychology, Stanford University, Stanford, CA, United States of America

\* knutson@stanford.edu

## Abstract

As social media becomes a key channel for news consumption and sharing, proliferating partisan and mainstream news sources must increasingly compete for users' attention. While affective qualities of news content may promote engagement, it is not clear whether news source bias influences affective content production or virality, or whether any differences have changed over time. We analyzed the sentiment of ~30 million posts (on twitter. com) from 182 U.S. news sources that ranged from extreme left to right bias over the course of a decade (2011–2020). Biased news sources (on both left and right) produced more high arousal negative affective content than balanced sources. High arousal negative content also increased reposting for biased versus balanced sources. The combination of increased prevalence and virality for high arousal negative affective content was not evident for other types of affective content. Over a decade, the virality of high arousal negative affective content also increased, particularly in balanced news sources, and in posts about politics. Together, these findings reveal that high arousal negative affective content may promote the spread of news from biased sources, and conversely imply that sentiment analysis tools might help social media users to counteract these trends.

## Introduction

Social media has changed how people consume and share news. Over half of adults in the United States report consuming at least some of their news online, either through news or social media sites (e.g., twitter.com, facebook.com) [1]. Social media news channels confer benefits, since the internet offers faster access to a broader range of content and platforms, which can promote sharing and discussion of information with others. These benefits may accompany costs, however, including the rapid dissemination and promotion of misinformation capable of sowing distress, division, and even death [2]. Thus, improved understanding of how news sources spread information and misinformation might eventually augment efforts to increase societal happiness, harmony, and health [3].

While some United States news sources offer more mainstream or balanced coverage, others feature more partisan or biased coverage. Concurrent with deepening political divisions, partisan news outlets have recently proliferated [4, 5]. Partisan news outlets are marked not only by subjective perceptions of bias from users, but also by objective evidence that they

**Data Availability Statement:** Summary data and analysis code supporting the reported findings are publicly available in an Open Science Framework repository (https://osf.io/63pzy/). Raw data, however, is owned by and available from x.com (formerly, twitter.com) through their Advanced

Programming Interface (https://developer.x.com/en/products/x-api). The authors initially used this API to access raw data (without special access).

**Funding:** This work was supported by NSF Grant 1732963, a Stanford Ethics, Society, and Technology Grant, and a Stanford HAI Faculty Seed Grant to JT and BK.

**Competing interests:** The authors have declared that no competing interests exist.

spread more misinformation [6, 7]. This misinformation may spread even faster and wider via news outlets' social media accounts. Compared with traditional media, social media platforms can confer virality through sharing, engagement, and comments [8, 9]. Thus, as news sources on social media grow, so may the spread of misinformation.

Social media news content can vary with respect to several attributes which can promote spread. Specifically, semantics (what was said) and source (who said it) can promote spread [10]. Beyond semantics and source, however, emerging evidence suggests that sentiment (or how it was said) can also command users' attention and engagement [5, 8, 9]. Which exact type of sentiment promotes the spread of online news content, however, remains unclear [11–13]. Further, research has not examined whether the impact of sentiment on news virality varies across a continuum of balanced versus biased news sources over time.

In this research, we sought to address two gaps in existing work. First, although some studies of online news affective content and virality have separately focused on either mainstream or partisan news sources [9], fewer have explicitly compared the two [14, 15]. Such a comparison is critical for understanding how partisan versus mainstream news outlets use sentiment to compete for users' attention online. While some studies of mainstream news outlets report that articles with high arousal affect (e.g., from *The New York Times*) are more likely to be reposted [11], other studies of partisan news content have reported that neither overall emotional strength [13] nor sensationalism (i.e., intentionally using extreme emotions and shock) predict engagement with news articles (e.g., on Facebook) [16]. One study found that negative emotional political messages were more likely to be reposted on social media, but did not distinguish between partisan and mainstream sources [17]. Another study reported that either positive or negative emotional language increased reposting about partisan topics within political ingroups (on Twitter), although the specific types of affect (e.g., positive or negative) which drove reposting depended on the topic [12]. To clarify and resolve these emerging findings, we directly compared affective content produced and spread by biased versus balanced news sources.

Second, while most research has focused on affect categories (e.g., positive versus negative sentiment, general emotional strength), more specific types of affective content that vary with respect to valence (positive or negative) as well as arousal (high or low arousal) may have distinct effects on the spread of content on social media. For example, some studies have reported an increased prevalence and virality of highly arousing negative emotional content (e.g., hate and outrage) relative to other types of emotional content in United States news content on social media [14, 18, 19] and online [20]. Another study demonstrated that misinformation (i.e., verified by fact-checking organizations) evoked more fearful, disgusted, and surprised reactions in social media users [9]. These findings imply that some types of negative aroused affective content may facilitate the faster and further spread of misinformation relative to information. Neuroeconomic research further implies that highly aroused positive and negative affect which typically occur during anticipation of uncertain outcomes promote subsequent attentional engagement and even action [21]. Since few studies have systematically compared all combinations of affective valence and arousal in social media news content, we examined whether the virality of affective content depends its valence and arousal. This approach [22] might help clarify inconsistent findings, while deepening mechanistic insights into which types of affective content are most prevalent and most powerfully drive engagement.

We addressed these questions by assessing which types of affective content were most prevalent and viral in social media posts of biased versus balanced news sources on social media [23]. To do so, we culled nearly 30 million posts from 182 U.S. news sources on a social media site (known as twitter.com during execution of this research) over the course of a decade (i.e.,

from 2011 to 2020). These brief posts typically contained headlines and summaries designed to entice users to read more extensive linked stories at affiliated news sites. News sources varied across a broad range of political orientations, from liberal/left-biased (e.g., *Wonkette*), to moderate/neutral (e.g., *PBS News Hour*), to conservative/right-biased (e.g., *Breitbart*) news sources. The social media platform under study (i.e., twitter.com) served as a news distribution platform and a forum for discussion, as well as a site of apparent ideological polarization [10] during this period. Using a modified version of the SentiStrength algorithm [19, 24], we quantified whether posts from these news sources contained four distinct types of affective content: (1) High Arousal Negative (or HAN; e.g., anxiety), (2) Low Arousal Negative (or LAN; e.g., dull), (3) Low Arousal Positive (or LAP; e.g., calm), and (4) High Arousal Positive (or HAP; e.g., excitement) affective content. Using this coded data, we addressed two key questions. First, does news source bias account for the prevalence of different types of affective content produced in posts? Second, is the affective content of a post associated with the extent to which it spreads (or is "viral" with respect to repost count), and do these associations vary for biased versus balanced news sources? We also explored whether these patterns generalized across time and news category (e.g., politics versus entertainment).

Given users' high engagement with biased news sources and the suspected virality of arousing affective content [14, 15], we predicted that biased news sources would produce more high arousal negative affective content, and that high arousal negative affective content would be more viral than other types of affective content. Given increasing partisan polarization in the United States [4, 5], we further predicted that the influence of news source bias on affective content prevalence and spread might increase over time and generalize across news categories.

## Materials and methods

### Data collection

We collected data from 215 news sources that varied with respect to political bias based on scores obtained by the non-partisan company Ad Fontes Media (adfontesmedia.com). Ad Fontes Media recruited individuals who vary across the political spectrum to rate a set of articles published by each news source for political bias using a highly-resolved scoring system (–42 = extreme left to 0 = neutral to +42 = extreme right). These scores were then aggregated to create a general political bias score for each news source [25]. In this study, we used scores from the adfontes.com website obtained on February 9, 2021.

Since this work focuses on political bias in the United States (U.S.), we excluded nineteen news sources that were not based in the U.S. and eight news sources that did not explicitly address politics. We also excluded two news sources that were cable versions of news sources already in the database (i.e., *CNN Cable TV News Network*, *Fox News TV Cable Network*), two news sources that lacked accounts on twitter.com (i.e., *Big League Politics*, *Rush Limbaugh*), and two news sources that had been officially banned from the platform for providing false content (i.e., *InfoWars*, *The Gateway Pundit*). Thus, the final dataset for analysis included a total of 182 news sources. The mean bias across these news sources leaned slightly to the political left (Mean = –1.75, SD = 14.67, range = –28.24 to 33.78).

To convergently validate news source bias scores, we extracted independent bias scores for an available subsample of overlapping news sources (n = 126) from allsides.com (accession date: Nov 20, 2021). These bias scores (ranging from 1 or leftmost to 5 or rightmost) were coded using a combination of editorial reviews, blind bias surveys, independent reviews, and third-party research. Despite differences in the derivation of measures, the correlation between Ad Fontes and AllSides bias scores was robust ($r(125) = 0.89$, $p < .001$; $\rho(125) = 0.88$, $p < .001$), supporting convergent validity of the news sources bias scores (Fig 1).

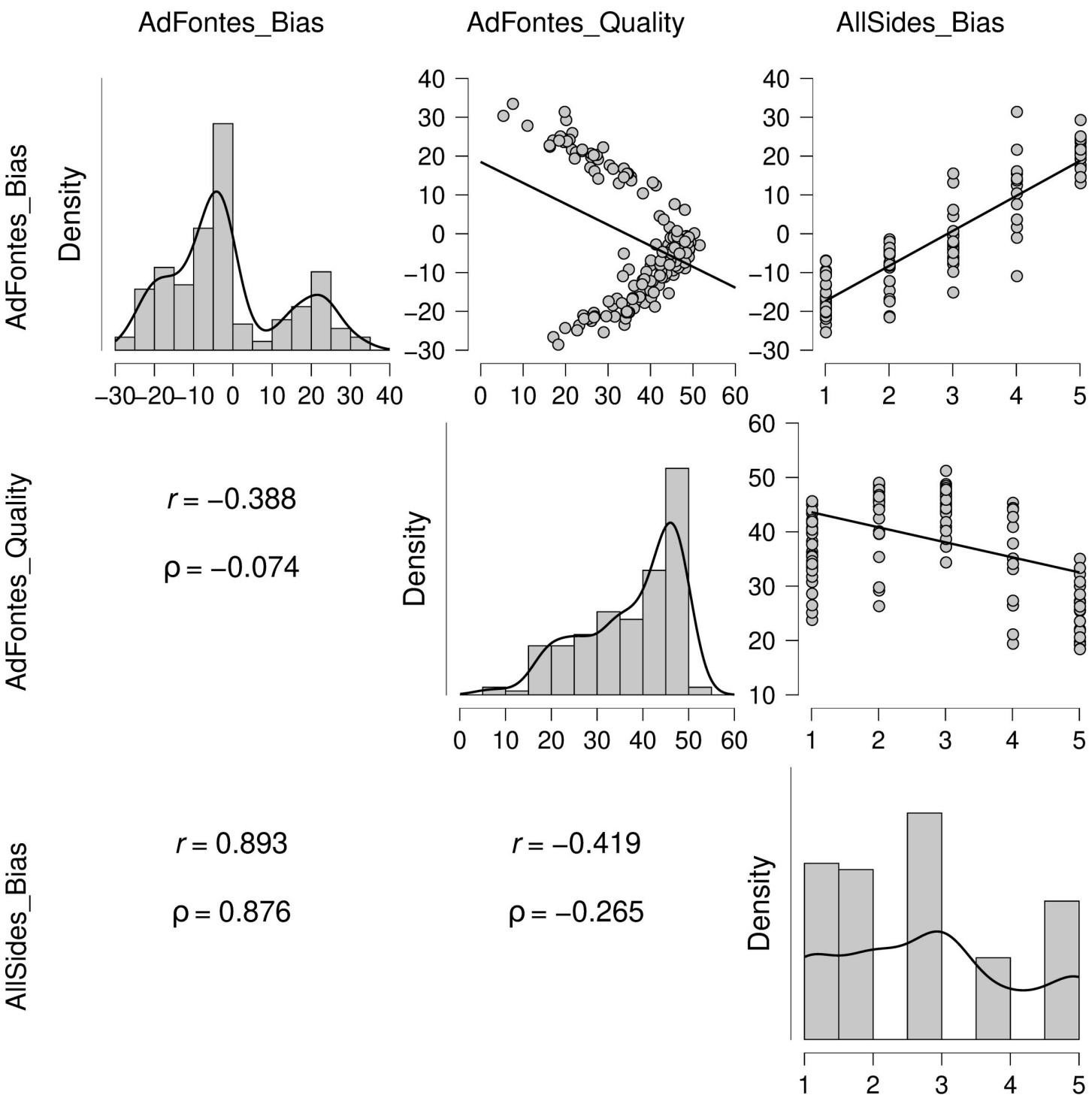

**Fig 1. Pairwise associations of AdFontes quality and bias scores with AllSides bias scores (parametric (*r*) and nonparametric (ρ) linear correlations; for 126 matched news sources from 182 total analyzed news sources).**

Using the Twitter Application Programming Interface (API) Academic Research Track, which allowed full-archive access during the study period, we collected posts posted by each news source on its official Twitter account over ten years (January 1, 2011 to December 31,

2020) to examine annual change. Data collection and processing complied with the terms and conditions imposed by twitter.com (now x.com). Because the Twitter repost (or "retweet") function was launched in late 2009, we excluded the year 2010 and began data collection in 2011 to ensure that the repost function had fully taken effect throughout the analyzed dataset. This yielded a total of 29,946,524 posts across 182 analyzed news sources.

## Sentiment analysis and categorization

To assess the affective content of message sentiment [19], we used a modified version of the SentiStrength algorithm [24]. SentiStrength relies on an English dictionary set that includes terms labeled with sentiment strength (valence and intensity), as well as semantically relevant terms including booster words (e.g., extremely, somewhat, quite), negating words (e.g. not, don't, couldn't), question words (e.g., why, how), emoticons (e.g., , (^ ^), ), slang words (e.g., lol, bff), and domain-specific terms (e.g., "must watch" in the context of film). The program then optimizes the term labels with machine learning, developed on a set of human-labeled social media web texts (i.e., six types of social media, including posts) [26] (Fig 2).

This version of the SentiStrength algorithm offered several advantages in coding sentiment relative to other algorithms. First, it is optimized to detect sentiment in short social media web texts and has been previously applied to content from Twitter [27] as well as news-related social media content [28]. Second, SentiStrength codes affective intensity separately from valence (i.e., each text is assigned a score of 1 = not positive to 5 = extremely positive and a score of 1 = not negative to 5 = extremely negative). Although intensity and arousal are not theoretically identical, they are often highly correlated in self-report measures, allowing SentiStrength to assume similarity. For instance, SentiStrength assigns calm and other low arousal positive states an intensity rating of 2, whereas it assigns excitement and other high-arousal positive states an intensity rating of at least 3. Further, exclamation marks receive higher intensity ratings. This version of the SentiStrength algorithm also includes updated emojis [19].

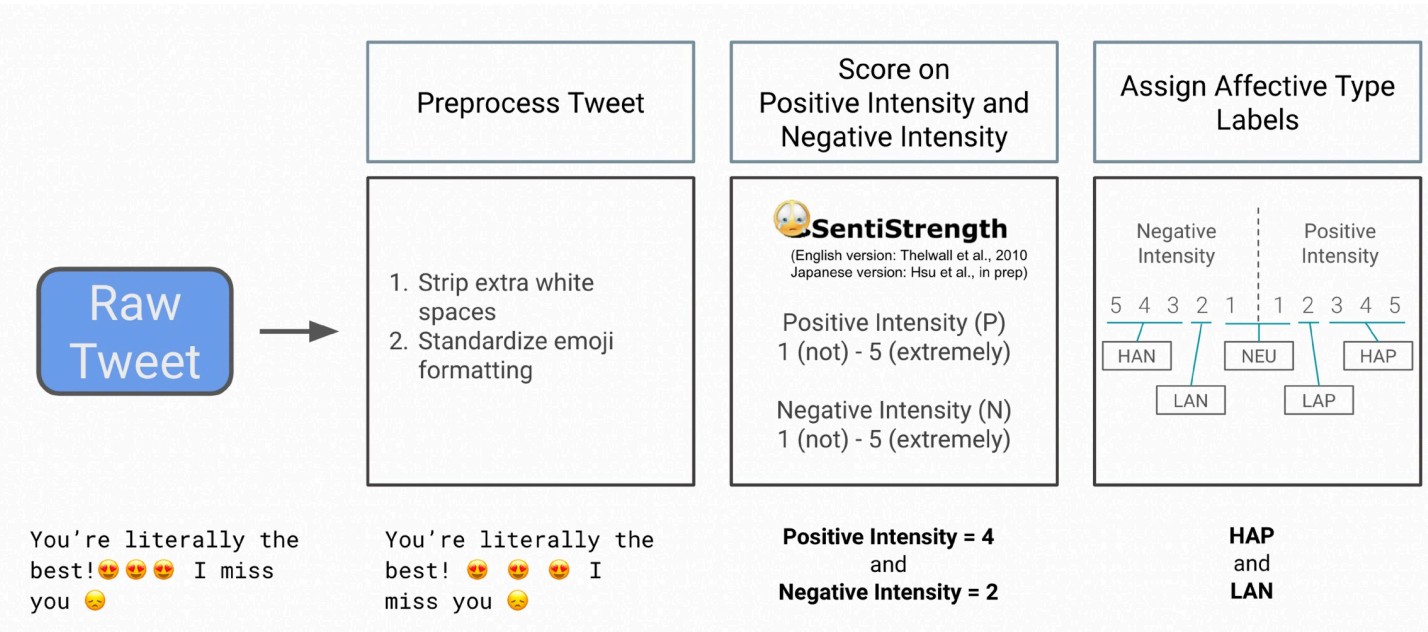

**Fig 2. Modified Sentistrength analysis flowchart (adapted from Hsu et al., 2021).** Raw posts were preprocessed, scored for positive and negative intensity, and assigned affective labels.

Posts were initially transformed to lower-case characters to support recognition of affective words in capitalized headlines.

To derive measures of affective content that varied in terms of both valence and arousal, posts were coded as containing: "Low Arousal Positive [LAP]" content if they received a SentiStrength positivity score of 2; "High Arousal Positive [HAP]" content if they received SentiStrength positivity scores of 3, 4, or 5.; "Low Arousal Negative [LAN]" content if they received SentiStrength negativity scores of 2; and "High Arousal Negative [HAN]" content if they received SentiStrength negativity scores of 3, 4, and 5. Since words psychometrically associated with "high arousal" (e.g., "excitement") were assigned a score of 3 by SentiStrength, we used scores of 3 and above to indicate high arousal affect. Thus, single posts could contain mixed affective content. Examples of posts containing different types of affect content include HAP (pos = 5, neg = 1): "'Jaw-droppingly exciting': Scientists discover seven Earth-like planets"; LAP (pos = 2, neg = 1): "Teaching kids to care, one book at a time"; LAN (pos = 1, neg = 2): "FDA reportedly planning stricter guidelines for release of COVID-19 vaccine"; and HAN (pos = 1, neg = 4): "Intruder shot to death with his own gun after barging into home, police say" (also see word clouds of post examples for each affect category in S1 Fig):

## Results

### Biased news sources produce different types of affective content

We first examined whether specific types of affective content were more prevalent overall. We calculated the overall prevalence of each affective type by first aggregating the percentage of posts that contained each type of affective content (%$Affect_k$) for each news source by year (e.g., the percentage of posts posted by ABC in 2020 that contained HAN affective content), then aggregating across years for each news source (e.g., the percentage of posts posted by ABC across the decade that contained HAN affective content), and finally aggregating across news sources. To compare different affect types, we fit mixed linear regression models using affect type to predict percentage with random intercepts of news source and year.

Analyses revealed that news sources posted more negative content (46.55%) than positive content (25.90%) overall (with 37.72% of the posts being classified as neutral). In terms of specific affect types, news sources posted low arousal negative content the most (LAN content: 25.84%), followed by high arousal negative content (HAN content: 20.71%), low arousal positive content (LAP content: 16.55%), and high arousal positive content (HAP content: 9.35%) (all pairwise comparisons between affect types were significant, Ps < .001). Thus, news sources posted almost twice as much negative as positive affective content overall (see S1 File for all regression specifications and results).

**Affective content prevalence by news source bias.** To test for an association of news source bias with affect prevalence, we calculated the prevalence of each type of affective content (%$Affect_k$) as described above for each news source aggregated across years. Then four linear models used the signed bias (*Bias*) and absolute bias (|*Bias*|) scores of each news source to predict the prevalence of each type of affective content, with a random intercept of year (i.e., one model for each k = {*HAN, LAN, LAP, HAP*}). We included the absolute value of political bias because political bias scores were bidirectional (i.e., negative = left-leaning, positive = right-leaning). The coefficient of the absolute bias score |*Bias*| quantified the effect of the magnitude of political bias in either direction on the prevalence of each affect type k (with a positive coefficient indicating that more political bias was associated with greater prevalence of affect type k). The coefficient of the signed bias score *Bias* tested whether bias direction mattered (with a positive coefficient indicating that more right-leaning political bias predicted greater prevalence

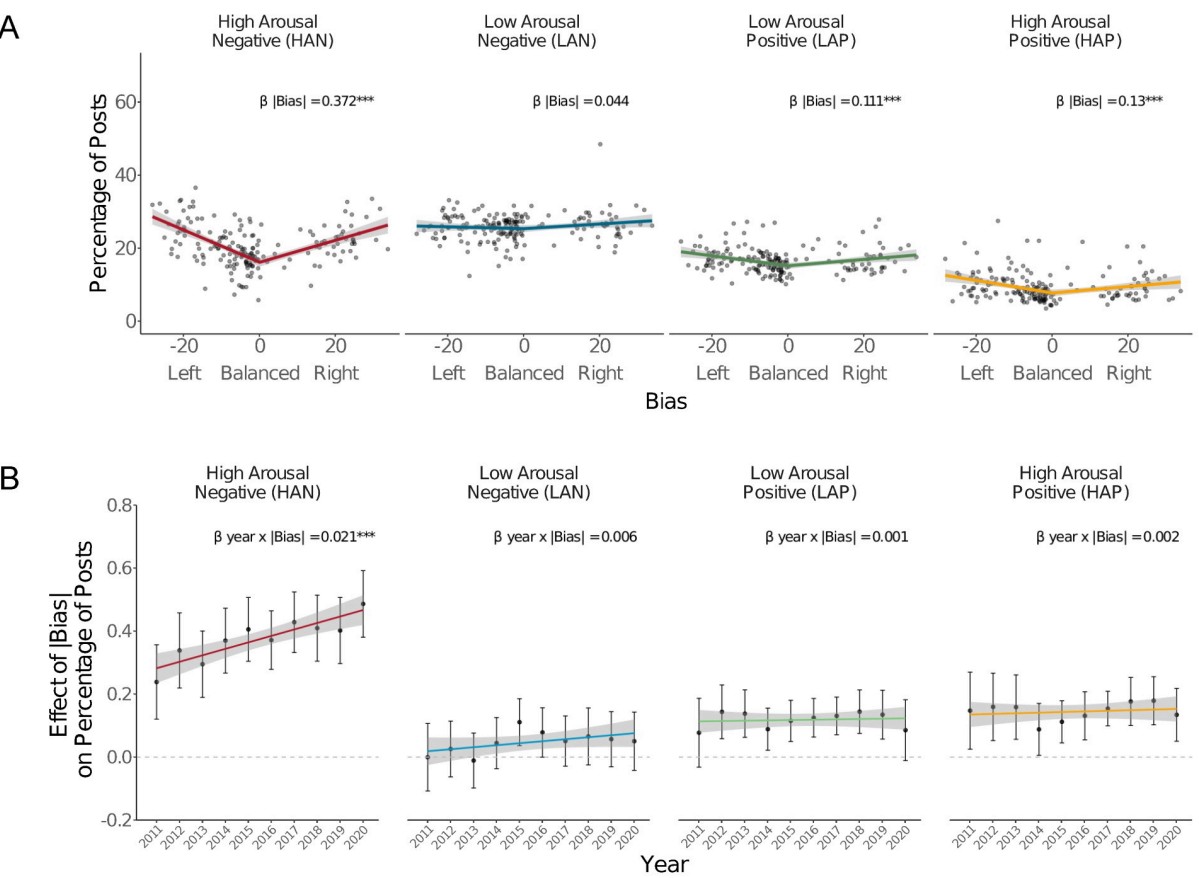

**Fig 3. Post affective content prevalence as a function of news source bias and year.** A) Effect of news source bias on the percentage of posts containing each affect type. Bands indicate 95% confidence intervals. Stars indicate significant coefficient for the absolute bias term in the model (** P < .01, *** P < .001). B) Effect of absolute political bias on the percentage of posts containing each affect type across years. Bars indicate 95% confidence intervals for linear fits. Stars indicate significant interaction between year and absolute bias (*** P < .001). Dashed lines represent the point at which absolute bias has no effect on the percentage of posts containing each affect type.

of affect type $k$, and a negative coefficient indicating that more left-leaning political bias predicted greater prevalence of affect type $k$).

Analyses revealed that absolute bias ($|Bias|$) of the news source had the largest effect on the prevalence of affective content. News sources that were more biased in either direction (i.e., higher in $|Bias|$) posted more HAN content. On average, a 1.0% increase in political bias predicted a 0.378% increase in the prevalence of HAN content ($b = 0.378$, SE = 0.017, P < 0.001) (Fig 3A). This association indicated that the most biased news sources in both directions (most extreme left bias = −28.24, most extreme right bias = 33.78) posted HAN content in approximately 11% more posts than the most balanced news sources. More biased news sources also posted more of other types of affective content, including LAN (0.05%; $b = 0.049$, SE = 0.014, P = 0.001), HAP (0.14%; $b = 0.145$, SE = 0.014, P < 0.001), and LAP content (0.12%; $b = 0.119$, SE = 0.012, P < 0.001). All these effects were smaller, however, than that observed for HAN content.

Analysis of signed political bias ($Bias$) indicated that more left-biased news sources posted more HAN (−0.07%; $b = −0.068$, SE = 0.001, P < 0.001), HAP (−0.04%; $b = −0.042$, SE = 0.008, P < 0.001), and LAP content (−0.02%; $b = −0.024$, SE = 0.007, P = 0.001), while more right-biased news sources posted more LAN content (0.02%; $b = 0.019$, SE = 0.008, P = 0.020), but

these effects were again smaller in magnitude (by a half or less) than the effect of absolute bias of news sources on the percentage of posts containing HAN content.

Thus, more biased news sources on both the left and right posted more high arousal negative (HAN) content. To a lesser extent, more biased news sources also posted more of other types of affective content (i.e., LAN, HAP, and LAP), relative to more balanced news sources.

**Affective content prevalence over time.**    Next, we examined whether these associations changed over time. Specifically, based on the association of high arousal negative affective content with political polarization in the United States [12, 15], we predicted that more politically biased news sources would post more HAN content over the course of a decade, but did not have predictions about other types of affective content.

To test for temporal trends, we modified the models above to replace the random intercept of year with a variable coded as *Year* since 2011 (year 2011 = 0, year 2012 = 1, . . ., year 2020 = 9) and interaction terms of Year with *Bias* and |*Bias*|. As predicted, a significant positive interaction between *Year* and |*Bias*| ($b = 0.020$, SE = 0.006, P < 0.001) emerged, indicating that more biased news sources posted more posts with HAN content over time (Fig 3B), relative to more balanced news sources. Importantly, this interaction was unique to HAN content, since the interaction of *Year* with |*Bias*| was not significant for the other three types of affective content (LAN: $b = 0.007$, SE = 0.005, P = 0.183; LAP: $b = 0.001$, SE = 0.004, P = 0.769; HAP: $b = 0.002$, SE = 0.005, P = 0.616). Thus, HAN content was the only type of affective content that biased news sources produced more of over time relative to balanced news sources. For signed political bias, there was an unpredicted and less significant positive interaction of *Year* with *Bias* and HAP content ($b = 0.008$, SE = 0.003, P = 0.004), suggesting that more right-biased news sources posted increasingly more HAP content over time.

These analyses revealed that more biased news sources posted more high arousal negative affective content compared to more balanced news sources over time (i.e., from 2011 to 2020; Fig 3A). Although overall more biased news sources had also posted more affective content of other types, those associations were less pronounced and did not change over time (Fig 3B).

## Biased news sources spread different types of affective content

Repost rates were standardized by calculating repost rate of a given post relative to other posts from the same news source within the same year. For example, if ABC posted 4 posts with repost counts 10, 20, 30, and 40 in year 2011, the post with repost count 30 would have a repost count percentile of 3 (its rank) / 4 (4 posts total) = 0.75. Thus, the indexed post would have been reposted equal to or greater than 75% of all posts posted by ABC in 2020.

We first examined whether posts with different types of affective content were reposted more often than others by fitting four linear models (one for each type of affective content $k$ = {*HAN, LAN, LAP, HAP*}) that used presence of each affect type (*Contains*$_k$ 0 = no, 1 = yes) to predict each post's repost count percentile. Post news source and posted year were modeled as random intercepts. As in previous research, whether the post contained media (e.g., pictures, videos) or not was also included as a covariate, since inclusion of media tends to increase spread [12, 15]. For each model, the estimate associated with predictor *Contains*$_k$ captured the average difference in repost count percentile of posts that contained $k$ versus posts that did not contain $k$. For example, if the estimate for HAN content was 2.00, posts containing HAN content were on average 2 percentiles higher in repost count than other posts.

**Affective content spread by news source bias.**    To examine whether these effects varied as a function of news source bias, we fit linear mixed effects models for each type of affective content $k$ = {*HAN, LAN, LAP, HAP*}. These predicted repost count percentile as a function of affect type (*Contains*$_k$ 0 = no, 1 = yes), news source bias (both signed *Bias* and absolute |*Bias*|),

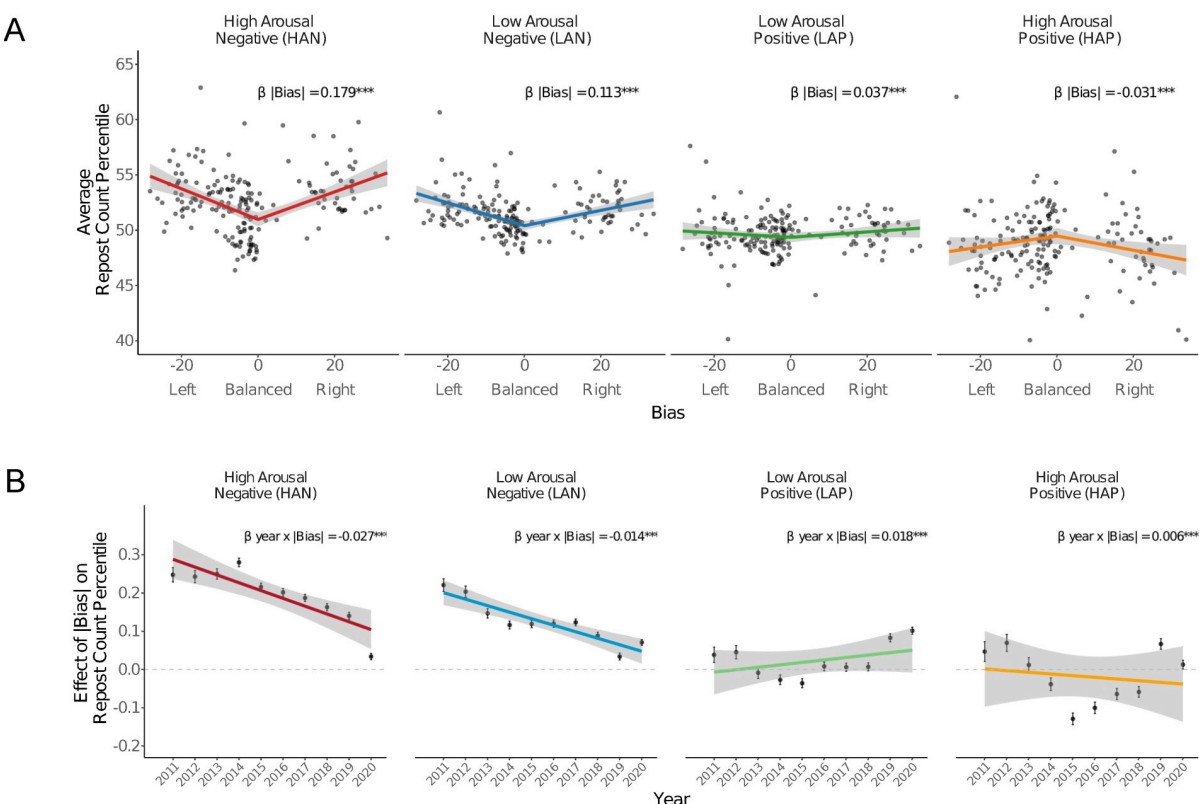

**Fig 4. Post affective content spread as a function of news source bias and year.** A) Effect of news source bias on the average repost count percentile of posts containing each affect type. Bands indicate 95% confidence intervals for linear fits. Stars indicate significant coefficient for the absolute value term in the model (*** P < .001). B) Effect of absolute political bias in repost count percentile posts containing each affect type across years. Bars indicate 95% confidence intervals. Stars indicate significant interaction between year and absolute bias (*** P < .001). Dashed lines represent the point at which absolute bias has no effect on the percentage of posts containing each affect type.

and their interaction ($Contains_k$ x $Bias$ + $Contains_k$ x $|Bias|$). Random intercepts of the news source and of each post's posted year were again included, along with the covariates described above. As in previous analyses, the signed coefficient represented the direction of bias, while the absolute coefficient represented the magnitude of bias.

Overall, posts that contained HAN content were reposted 3.27 percentiles more than posts that lacked HAN content ($b$ = 3.272, SE = 0.013, P < .001), consistent with previous findings demonstrating a viral effect of HAN content in United States users (Fig 4) [19]. As predicted, posts with HAN content were reposted more for biased than for balanced news sources. Specifically, while posts containing HAN content were reposted 1.46 percentiles more among balanced news sources (simple effect of HAN at bias = 0 $b$ = 1.457, SE = 0.022, P < .001), they were reposted by an additional 0.179 percentiles with each percent of increase in bias magnitude ($Contains_{HAN}$ x $|Bias|$ interaction: $b$ = 0.179, SE = 0.002, P < .001). Also, with each percent decrease in signed bias (i.e., more leftward), posts containing HAN content were reposted by an additional 0.009 percentiles ($Contains_{HAN}$ x $Bias$ interaction $b$ = –0.009, SE = 0.001, P < .001). Together, these effects imply that for the leftmost sources (bias = –28.24), posts containing HAN content were reposted 5.30 percentiles more than posts without HAN content, and for the rightmost sources (bias = 33.78), posts containing HAN content were reposted 5.73 percentiles more than posts without HAN content (Fig 4).

Analyses also revealed some unexpected patterns for other types of affective content. For instance, posts containing LAN content were reposted 0.698 percentiles more than posts without LAN content ($b$ = 0.698, SE = 0.020, P < .001), and these were also reposted more for biased news sources ($Contains_{LAN}$ x $|Bias|$ interaction: $b$ = 0.114, SE = 0.002, P < .001). While similar to the effect of HAN content on reposting, the effect of LAN content on reposting was smaller, and more prominent for biased sources on the left ($Contains_{LAN}$ x $Bias$ interaction $b$ = −0.028, SE = 0.001, P < .001). In contrast, posts containing LAP and HAP content were reposted less than posts containing other content (LAP: $b$ = −1.340, SE = 0.024, P < .001; HAP: $b$ = −1.600, SE = 0.032, P < .001). Posts containing LAP content were reposted more by biased media sources, particularly on the right ($Contains_{LAP}$ x $|Bias|$ interaction: $b$ = 0.037, SE = 0.002, P < .001; $Contains_{LAP}$ x $Bias$ interaction: $b$ = 0.008, SE = 0.001, P < .001). This pattern of findings might cohere with previous results suggesting that partisan news can trigger positive emotions supporting the ingroup to boost political engagement [29]. Posts containing HAP content, however, were instead reposted less by biased news sources ($Contains_{HAP}$ x $|Bias|$ interaction: $b$ = −0.031, SE = 0.003, P < .001; $Contains_{HAP}$ x $Bias$ interaction: $b$ = 0.019, SE = 0.002, P < .001).

Together, these findings supported predictions that posts containing HAN content would be reposted more than posts containing other types of affective content, and that this effect would be stronger for more biased news sources. Consistent with predictions, more biased news sources not only posted more HAN content, but also showed greater spread for posts containing HAN content. A similar but weaker pattern was apparent for posts containing LAN content.

**Affective content spread over time.** Based on the previously-documented association of negative emotional content with political polarization [12, 15, 18], we predicted that the effect of news source bias on reposts of posts containing HAN content might increase over time, but did not have specific predictions for other affect types.

To examine spread over time, we modified models of affective content to include a *Year* variable (coded as year since 2011: year 2011 = 0, year 2012 = 1, . . ., year 2020 = 9) that generated three-way interactions (with $Contains_k$ x $Bias$ and $Contains_k$ x $|Bias|$). We therefore predicted a significant positive interaction for *Year* x $Contains_{HAN}$ x $|Bias|$. While this interaction was significant, contrary to prediction, its coefficient weight was *negative* rather than positive (*Year* x $Contains_{HAN}$ x $|Bias|$ interaction: $b$ = −0.027, SE = 0.000, P < .001), implying that the effect of news source bias on the repost count percentile of posts containing HAN content *decreased* over years. A similar but smaller interaction was also apparent for LAN content (*Year* x $Contains_{LAN}$ x $|Bias|$ interaction: $b$ = −0.014, SE = 0.001, P < .001). This interaction was significantly positive for both LAP and HAP content, however, suggesting that the effect of news source bias on the repost count percentile of posts containing LAP or HAP content increased over years (LAP: *Year* x $Contains_{LAP}$ x $|Bias|$ interaction: $b$ = 0.018, SE = 0.001, P < .001; HAP: *Year* x $Contains_{HAP}$ x $|Bias|$ interaction: $b$ = 0.006, SE = 0.001, P < .001).

This negative effect of news bias on reposts of posts containing HAN content over time did not fit predictions. According to one account, reposts of posts containing HAN content might have decreased for biased news sources over time. According to an alternative account, however, reposts of posts containing HAN content instead might have instead increased for balanced news sources over time. To distinguish between these accounts, we split news sources into more "balanced" versus "biased" sets using a split on the median $|Bias|$ score (i.e., 10.67). Next, we fit linear mixed effects models that predicted repost count percentile as a function of whether the post contained HAN content ($Contains_{HAN}$ 0 = no, 1 = yes), year (coded as year since 2011: year 2011 = 0, year 2012 = 1, . . ., year 2020 = 9), and the interaction of affect

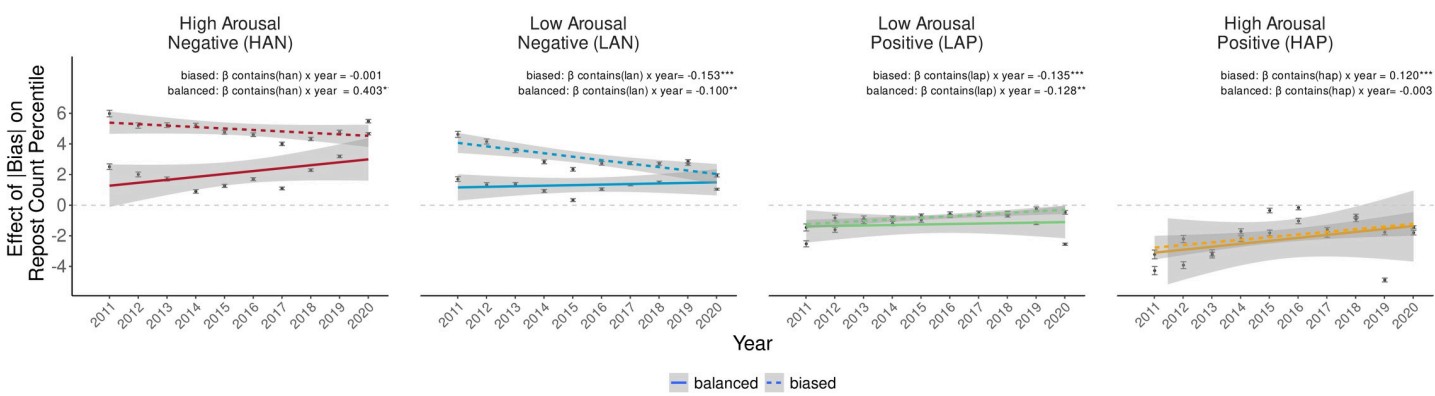

**Fig 5. Post affective content spread as a function of year grouped by biased (dashed) versus balanced (solid) news sources (split by median of absolute bias).** Bars indicate 95% confidence intervals for linear fits. Stars indicate significant interaction between year and absolute bias (*** P < .001). Gray dashed lines represent the point at which absolute bias has no effect on the percentage of posts containing each affect type.

content with year. Random intercepts for news sources were included, along with the covariates described above (Fig 4B).

Findings supported the second account, since the effect of HAN content on repost count percentile did not change over years for more biased news sources, but instead *increased* over years for more balanced news sources (biased: *Year* x *Contains*$_{HAN}$: $b = -0.001$, SE = 0.008, P = 0.896; balanced: *Year* x *Contains*$_{HAN}$: $b = 0.402$, SE = 0.007, P < .001) (Fig 4). For other affect types, LAN content also spread less for more biased news sources, but spread more for more balanced news sources (biased: *Year* x *Contains*$_{LAN}$: $b = -0.153$, SE = 0.008, P < .001; balanced: *Year* x *Contains*$_{LAN}$: $b = 0.100$, SE = 0.006, P < .001). Opposite patterns emerged, however, for the remaining affect types. Specifically, LAP content spread more for more biased news sources, but spread less for more balanced news sources (biased: *Year* x *Contains*$_{LAP}$: $b = 0.135$, SE = 0.009, P < .001; balanced: *Year* x *Contains*$_{LAP}$: $b = -0.128$, SE = 0.007, P < .001), and HAP content also spread more for more biased news sources but did not change over time for more balanced news sources (biased: *Year* x *Contains*$_{HAP}$: $b = 0.120$, SE = 0.011, P < .001; balanced: *Year* x *Contains*$_{HAP}$: $b = -0.003$, SE = 0.009, P = 0.723) (Fig 5).

**Affect prevalence and spread by news category.** To examine whether the association of news source bias with prevalence and spread of high arousal negative content generalized across different types of news, we modeled news content topics with a Latent Dirichlet Allocation (LDA) analysis [30]. To create semantically relevant topics, we incorporated priors into the LDA optimization by initializing topic distribution parameters with estimates from human coders. Specifically, we first divided the dataset into a smaller training sample of post hashtags that appeared most frequently (i.e., those that were ranked in the top 500 hashtags with respect to frequency). Next, we identified seven news categories along with short descriptions for coding their labels (i.e., "politics and policy," "sports," "health and wellness," "business and economy," "crime and safety," "culture and lifestyle," and "science and technology," see S1 File for category label descriptions). Two trained raters then classified the 500 highest-frequency hashtag terms into these categories. The trained raters showed high inter-rater reliability (kappa = .743) in their categorical classifications of these hashtag terms. Labeled hashtags were then used to estimate the topic distribution of the training sample and the document-topic distribution in the LDA model. The LDA model was then optimized from the labelled training sample (using the Gensim 4.1.2 Python package) and generalized to the remainder of the dataset. This model then assigned the most likely news category label to each post, retaining label predictions that exceeded a 1% false positive rate.

Out of a total of 10,315,511 labeled posts, assigned percentages varied by label category (i.e., with the model assigning 51.0% to "politics and policy," 15.7% to "sports," 10.5% to "health and wellness," 7.0% to "business and economics," 6.3% to "crime and safety," 5.7% to "culture and lifestyle," and 3.8% to "science and technology"). Analyses of affective content and spread implemented for the entire sample of posts were then repeated specifically for posts within each of these categories. The association of news source bias with more prevalent as well as more viral high arousal negative affective content was most apparent in the largest news category of "politics and policy" (i.e., > 50% of posts) and less so in "business and economics", but not in the other news categories (i.e., "sports," "health and wellness," "crime and safety," "culture and lifestyle," or "science and technology"; see S1 File for plotted LDA results).

## Discussion

Sentiment analysis of nearly 30 million social media posts from 182 news sources varying in partisan bias over the course of a decade (January 1, 2011 to December 31, 2020) revealed that biased news sources produced more highly arousing negative affective content. Specifically, the most biased news sources (on both the left and the right) included approximately 12% more high arousal negative affective content in their posts than balanced news sources, and this trend increased over time. Further, posts that contained highly arousing negative affective content were more reposted, spreading more than those that did not. Critically, this virality effect was stronger for more biased news sources, since highly arousing negative affective content increased sharing (or reposts) by 5.30% in extreme left sources and by 5.73% in extreme right sources, compared to 1.46% in more balanced news sources. Unexpectedly, this difference between spread of high arousal negative content in biased versus balanced news sources appeared to diminish over time. Further analyses revealed, however, that this was not due to decreasing spread of high arousal negative content in biased news sources, but rather to increasing spread of high arousal negative content in balanced news sources over time.

The combination of increased prevalence and increased spread was specific to high arousal negative affective news content, but not other types of affective content. Although posts with low arousal negative affective content were also reposted more in biased news sources, biased news sources did not post more low arousal negative affective content than balanced news sources, and the association of low arousal negative content with spread was smaller than that for high arousal negative content. Biased news sources also posted more low arousal positive affective content and high arousal positive affective content than balanced news sources, but these effects were again smaller than for high arousal negative affective content, and posts containing low arousal positive affective content and high arousal positive affective content did differentially spread from biased versus balanced news sources. Exploratory analyses suggested that this pattern of higher prevalence and spread of high arousal negative content was primarily present in posts related to politics and policy, which comprised over half of the analyzed news posts (see S1 File).

This work provides an initial overview of the prevalence and spread of different types of affective content in social media posts by United States news sources that vary along a continuum of partisan bias over the course of a decade. By examining affective content that included all combinations of valence and arousal, we were able to isolate high arousal negative affective content as uniquely more prevalent and more viral in posts from biased news sources. This distinction between high and low arousal affect allowed us to test a prediction derived from an anticipatory affect model that highly arousing affective content would be more likely to capture attention and motivate behavior [21]. In the current findings, however, only high arousal negative affect showed a clear association with subsequent spread, extending previous findings

associating negative affective content with news virality [15]. Notably, these patterns did not depend on the direction of news source bias, instead extending to biased news sources on both the left and the right [14]. These results add specificity to emerging findings which imply that negative affective content predominates and can drive virality in United States news media on both ends of the political spectrum [9, 12, 14, 15]. The prevalence and spread of high arousal negative content in biased sources was not limited to current developments or events, since it generally increased over ten years. Further, the virality of high arousal negative content did not decrease over time for biased news sources, but rather increased over time for balanced news sources.

Despite the fact that individual American social media users tend to produce more positive than negative affective content (i.e., ~50% more positive than negative affective content on twitter.com) [19], news sources instead produced more negative than positive affective content overall (i.e., ~80% more negative than positive content on twitter). This asymmetry in the production of affective content implies that part of the high arousal negative content circulating on social media may emanate not only from individual users, but even more from biased news media. This pattern of findings is also consistent with reports that social media algorithms may boost high arousal negative content [31]. Specifically, ranking algorithms on many social media platforms may learn from users' engagement behavior (e.g., reposts) to prioritize and prominently present engaging content [32]. This algorithmic boost might create feedback loops that amplify the spread and reach of that engaging content. Since high arousal negative posts appear to elicit greater engagement, these posts may receive higher ranking in social media timelines, and so reach more users. To further maximize engagement in an increasingly competitive media landscape, news outlets might therefore be incentivized to produce even more high arousal negative affective content. Effects of such a feedback loop might then intensify in biased news sources since they post relatively more high arousal negative affective content and tend to show greater spread for those posts.

Excessive high arousal negative content might therefore generate a form of "affective pollution" by imposing unwanted costs on individual users, since social media users in the United States behaviorally appear to prefer producing positive content when left to their own devices [19]. Costs of this "affective pollution" could take several forms. First, negative arousing content might increase negative affect and decrease well-being in users, either over the short or the long term [33]. Second, and based on its prevalence in biased media, negative arousing content might boost attention to and subsequent acceptance of associated misinformation versus information [33, 34]. Third, and particularly in for outgroup targets, negative arousing content may increase political polarization and hostility towards others [15]. Thus, although negative affective content prevalence combined with spread may increase engagement and associated income for social media platforms [8, 9, 12], it may also impose a toxic downward spiral of unwanted negative costs on individual users.

Researchers have posited that the virality of high arousal negative affect on social media might be due to the salience of threat [12], an absence of channels for corrective feedback (e.g., a lack of retaliation for norm-violating behavior; [18]), or the surprisingness of affective content that violates cultural values [19]. Future research might examine the influence of these mechanisms on content from biased versus balanced news sources. This research focused on available data from only one social media platform (i.e., the former twitter.com). Another recent study of headline sentiment culled from a subset of these news sources across platforms on the open internet, however, reached similar conclusions about a general rise in negative emotional content from 2000–2020 [20]. Since the present research focused on news from the United States in social media, generalizability to other cultures also deserves investigation. For instance, the viral influence of high arousal negative affective content might not generalize to

more interdependent East Asian cultures and so may fail to elicit similar engagement [19]. Comparing news bias and affective content in social media across cultures might also help to elaborate and refine global efforts to curb misinformation.

Future research might characterize the impact of affective content on other social media platforms and involving different groups in greater detail. With respect to different social media platforms, while the previous open accessibility of data (e.g., on twitter.com) made this large-scale analysis possible, generalizability should ideally be established to other closed social media platforms. Consistent with the present findings, a similar influence of negative affective content on post virality has been noted on at least one other prominent closed social media platform (i.e., Facebook) [15]. With respect to group identity, researchers could analyze whether affective content and spread varies as a function of the topic under discussion–particularly with respect to polarizing topics that denigrate outgroups or elevate the ingroup [15]. Consistent with a group identity account, exploratory analyses implied that the influence of negative affective content on virality was most pronounced for political news, which constituted the majority of analyzed posts. To a lesser extent than high arousal negative affective content, posts containing low arousal positive affective content also spread more in biased news sources. Further research might test whether virality is associated with positive affective content supporting the ingroup [29]. While this research focused on total reposts, future analyses might probe how affective content influences reposting across different user networks, both within and between different groups [12].

These findings coincide with a rise in scientific research on affective phenomena due to advances in measurement (including sentiment analysis), which allow researchers to test qualitative predictions with quantitative tools [35]. Recently, researchers have developed even more precise and granular tools for sentiment analysis (e.g., Large Language Models), which might be leveraged to test the specificity as well as the generalizability of these findings [36]. Measurement and quantification of affective content further implies novel possibilities for intervention. Beyond characterizing the prevalence and spread of affective content, precise interventions might be designed that can reduce negative aroused affective sentiment in an unbiased manner (i.e., independent of message source and semantics). The present findings suggest that interventions with the capacity to identify and filter negative aroused affective content might also decrease engagement with biased news content, possibly shifting user exposure towards more balanced and veridical news content. Since previous research suggests that American social media users preferentially produce positive rather than negative content, negative arousing content may "hijack" users' preferences, leading to unintended and unwanted decreases in well-being, information quality, and social harmony. Digital literacy programs might therefore refer to the current findings to educate users about how their affect can be disrupted by biased news sources. Understanding how affective content influences engagement might also inform efforts to design more optimal algorithms that diminish rather than amplify the spread of biased news. Such algorithms might provide companies, governments, and especially individual users with tools capable of countering the rapid spread of damaging and divisive online content. Since this research focused on naturalistic data, we could only investigate mechanisms that might drive the observed effects, leaving the potential impact of relevant causal interventions open for future exploration and development [37, 38].

## Supporting information

**S1 Fig. Word prevalence clouds by post affect type (derived from 700 randomly-selected example posts per affect type category).**
(TIF)

**S1 File. Supporting methods and results.**
(DOCX)

## Acknowledgments

We thank Alexis Ceja for assistance with classification, Nicole Deng for assistance with analysis, and Yiwei Luo, Jonas Schöne, the Tsai lab, Spanlab, and two anonymous reviewers for feedback on previous drafts.

## Author Contributions

**Conceptualization:** Brian Knutson, Tiffany W. Hsu, Jeanne L. Tsai.

**Data curation:** Brian Knutson.

**Formal analysis:** Brian Knutson, Tiffany W. Hsu, Michael Ko, Jeanne L. Tsai.

**Funding acquisition:** Brian Knutson, Jeanne L. Tsai.

**Investigation:** Brian Knutson, Tiffany W. Hsu, Michael Ko.

**Methodology:** Brian Knutson, Tiffany W. Hsu, Michael Ko, Jeanne L. Tsai.

**Project administration:** Brian Knutson, Jeanne L. Tsai.

**Supervision:** Brian Knutson, Jeanne L. Tsai.

**Visualization:** Brian Knutson.

**Writing – original draft:** Brian Knutson, Tiffany W. Hsu, Jeanne L. Tsai.

**Writing – review & editing:** Brian Knutson, Tiffany W. Hsu, Michael Ko, Jeanne L. Tsai.

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
