## [Decision Letter · Decision Letter 0]

23 Jul 2024

PONE-D-24-21176News source bias and sentiment on social mediaPLOS ONE

Dear Dr. Knutson,

Thank you for submitting your manuscript to PLOS ONE. After careful consideration, we feel that it has merit but does not fully meet PLOS ONE’s publication criteria as it currently stands. Therefore, we invite you to submit a revised version of the manuscript that addresses the points raised during the review process.While the two experts who reviewed your manuscript considered it to be strong and have great potential for publication, they also made suggestions regarding the contextualization and the illustration of your assessment and the verification of some methodological issues which I believe will help to improve your paper. 

We look forward to receiving your revised manuscript.

Kind regards,

Silvio Eduardo Alvarez Candido

Academic Editor

PLOS ONE

2. (1) In your Methods section, please include additional information about your dataset and ensure that you have included a statement specifying whether the collection and analysis method complied with the terms and conditions for the source of the data.

(2) Please note that PLOS ONE has specific guidelines on code sharing for submissions in which author-generated code underpins the findings in the manuscript. In these cases, we expect all author-generated code to be made available without restrictions upon publication of the work. Please review our guidelines at https://journals.plos.org/plosone/s/materials-and-software-sharing#loc-sharing-code and ensure that your code is shared in a way that follows best practice and facilitates reproducibility and reuse.

 [This work was supported by NSF Grant 1732963, a Stanford Ethics, Society, and Technology Grant, and a Stanford HAI Faculty Seed Grant to JT and BK. ].  

[We thank Alexis Ceja for assistance with coding and classification, as well as Yiwei Luo, Jonas Schöne, the Tsai lab, and Spanlab for feedback on previous drafts. This work was supported by NSF Grant 1732963, a Stanford Ethics, Society, and Technology Grant, and a Stanford HAI Faculty Seed Grant to JT and BK. ]

 [This work was supported by NSF Grant 1732963, a Stanford Ethics, Society, and Technology Grant, and a Stanford HAI Faculty Seed Grant to JT and BK. ]

6. We note that you have indicated that there are restrictions to data sharing for this study. PLOS only allows data to be available upon request if there are legal or ethical restrictions on sharing data publicly. For more information on unacceptable data access restrictions, please see http://journals.plos.org/plosone/s/data-availability#loc-unacceptable-data-access-restrictions. 

7. Please remove your figures from within your manuscript file, leaving only the individual TIFF/EPS image files, uploaded separately. These will be automatically included in the reviewers’ PDF.

Reviewers' comments:

Reviewer's Responses to Questions

**Comments to the Author**

1. Is the manuscript technically sound, and do the data support the conclusions?

Reviewer #1: Yes

Reviewer #2: Yes

2. Has the statistical analysis been performed appropriately and rigorously? 

Reviewer #1: Yes

Reviewer #2: Yes

3. Have the authors made all data underlying the findings in their manuscript fully available?

Reviewer #1: No

Reviewer #2: Yes

4. Is the manuscript presented in an intelligible fashion and written in standard English?

Reviewer #1: Yes

Reviewer #2: Yes

5. Review Comments to the Author

Reviewer #1: This work investigates whether the bias of the news source affects the creation of emotional content and its spread, and whether these differences have evolved over time. The authors analyzed the sentiment of 30 million tweets from 182 U.S. news sources, ranging from extreme left to extreme right bias, over a decade (2011-2020). The results indicate that biased news sources, whether left or right, produced more high-arousal negative emotional content compared to balanced sources. This high-arousal negative content also increased the likelihood of reposting for biased sources as opposed to balanced ones (but this trend did not hold for other types of emotional content). In recent years, the virality of high-arousal negative content rose, especially in balanced news sources and posts about politics.

I would like to commend the authors for a methodologically robust work. The study is well thought out. The results are perhaps not too surprising and mostly consistent with the pre-existing literature but it is still important to do the quantitative documentation of such trends. The attention to detail and the robust methodology is visible in many places of the paper like for instance when the authors triangulate their initial choice of news media bias ratings (from Ad Fontes) with another provider (AllSides). The LDA topic modeling to characterize the news sources sections driving the results was another nice touch. The statistical analysis is solid and some of the somehow counterintuitive results (like the increased high arousal negative affective content in balanced news sources) are an important contribution to the existing literature. I barely have much criticism to utter and my recommendation is to accept this paper with only minor revisions.

I would also like to commend again the authors because as a reader of their manuscript they come across as very impartial in their work, as it should be!. This is unfortunately not always the case when reading these sorts of paper, and it feels like some authors have an agenda or some sort of viewpoint bias that they try to accommodate the results to. This is not the case at all in the current paper. It feels exquisitely neutral in that regard.

Only a bit of minor constructive criticism for the authors at the risk of being pedantic. I think the weakest link in the paper are the visualizations where there is definitely room for improvement. All the detailed reporting about the different linear models used for the analysis in the text is fine but sometimes a compelling visualization can convey the point across very efficiently.

The comments in the discussion section about the spread of negative affect news content through the population is highly relevant and important. Perhaps the authors should mention at least a sort of counter hypothesis: that people perhaps could be developing some sort of resistance or immunity after perhaps too much exposure to negative sentiment in their news diets and could be tuning out. I'm not saying that this is happening, but perhaps the authors should concede this as one of many potential hypotheses.

Finally, the usage of SentiStrength is well reasoned. I was just wondering whether an LLM like gpt-3.5-turbo would have a high degree of interrater agreement with the SentiStrength algorithm annotations, thus providing further evidence for the validity of the findings. But again, this is a bit pedantic.

To conclude, well done to the authors for a solid piece of academic work.

Reviewer #2: The paper contributes to fill empirical gaps though solid assessments. It deals with a very important issue of how political bias influence the diffusion of news. The findings are not so surprising, but clearly generate an empirical contribution. I have only a couple of suggestions to improve the work after publication.

While the empirical analysis of the paper is excellent, the paper is a bit atheoretical to my taste. I miss recourse to more general ideas and mechanisms in the contextualization regarding existing explanations about why would biased theory spread more. In my view this would be important to better contextualize the paper in nearby discussions. This would be particularly important for the introduction of the paper.

Another point refers to the illustration of the findings. First, the definition of some figures may be improved. Also, in some parts the authors describe textually findings that could be nicely visually represented.

I congratulate the authors for the excellent work.

6. PLOS authors have the option to publish the peer review history of their article (what does this mean?). If published, this will include your full peer review and any attached files.

Reviewer #1: No

Reviewer #2: No

---

## [Author Response · Author response to Decision Letter 0]

10 Sep 2024

We have now updated our style to meet the PLOS ONE style requirements throughout. 

2. (1) In your Methods section, please include additional information about your dataset and ensure that you have included a statement specifying whether the collection and analysis method complied with the terms and conditions for the source of the data.

We now state in the Methods that: “Data collection and processing complied with the terms and conditions imposed by twitter.com (now x.com).” 

(2) Please note that PLOS ONE has specific guidelines on code sharing for submissions in which author-generated code underpins the findings in the manuscript. In these cases, we expect all author-generated code to be made available without restrictions upon publication of the work. Please review our guidelines at https://journals.plos.org/plosone/s/materials-and-software-sharing#loc-sharing-code and ensure that your code is shared in a way that follows best practice and facilitates reproducibility and reuse.

We now provide our analysis code at: https://osf.io/63pzy/. 

We have removed the updated funding information from the acknowledgments and now provide it in response to item 5 below.

 [This work was supported by NSF Grant 1732963, a Stanford Ethics, Society, and Technology Grant, and a Stanford HAI Faculty Seed Grant to JT and BK.]. 

We now state in the Financial Disclosure that: “The authors have no conflicts of interest to declare. The funders had no role in study design, data collection and analysis, decision to publish, or preparation of the manuscript.” (line 587)

[We thank Alexis Ceja for assistance with coding and classification, as well as Yiwei Luo, Jonas Schöne, the Tsai lab, and Spanlab for feedback on previous drafts. This work was supported by NSF Grant 1732963, a Stanford Ethics, Society, and Technology Grant, and a Stanford HAI Faculty Seed Grant to JT and BK.]

 [This work was supported by NSF Grant 1732963, a Stanford Ethics, Society, and Technology Grant, and a Stanford HAI Faculty Seed Grant to JT and BK.]

Thank you for this clarification -- we have removed this information from the Acknowledgments section and would like the Funding Statement to read, and now also mention this in our cover letter: 

“This work was supported by NSF Grant 1732963, a Stanford Ethics, Society, and Technology Grant, and a Stanford HAI Faculty Seed Grant to JT and BK. The funders had no role in study design, data collection and analysis, decision to publish, or preparation of the manuscript.”

6. We note that you have indicated that there are restrictions to data sharing for this study. PLOS only allows data to be available upon request if there are legal or ethical restrictions on sharing data publicly. For more information on unacceptable data access restrictions, please see http://journals.plos.org/plosone/s/data-availability#loc-unacceptable-data-access-restrictions. 

We now provide summary data, but not raw data, which is prohibited by the current x.com data sharing policy. We also note this in the Data Availability statement (see below).

We now provide summary data in an OSF repository, as noted in the Data Availability statement: 

“Summary data and analysis code supporting the reported findings are publicly available in an Open Science Framework repository (https://osf.io/63pzy/). Restrictions now apply to the raw data based on the data policy of x.com (formerly, twitter.com), which were used under license and so are no longer publicly available.” (line 591)

7. Please remove your figures from within your manuscript file, leaving only the individual TIFF/EPS image files, uploaded separately. These will be automatically included in the reviewers’ PDF.

We have removed the figures from the manuscript file and uploaded them separately.

We now include captions for our Supporting Information files at the end of the manuscript and have updated in-text citations accordingly. 

We have reviewed and revised our references section to conform to PLOS ONE style. 

Reviewers' comments:

Reviewer's Responses to Questions

Comments to the Author

1. Is the manuscript technically sound, and do the data support the conclusions?

Reviewer #1: Yes

Reviewer #2: Yes

2. Has the statistical analysis been performed appropriately and rigorously? 

Reviewer #1: Yes

Reviewer #2: Yes

3. Have the authors made all data underlying the findings in their manuscript fully available?

Reviewer #1: No

Reviewer #2: Yes

4. Is the manuscript presented in an intelligible fashion and written in standard English?

Reviewer #1: Yes

Reviewer #2: Yes

5. Review Comments to the Author

Responses to Reviewers

Reviewer #1: This work investigates whether the bias of the news source affects the creation of emotional content and its spread, and whether these differences have evolved over time. The authors analyzed the sentiment of 30 million tweets from 182 U.S. news sources, ranging from extreme left to extreme right bias, over a decade (2011-2020). The results indicate that biased news sources, whether left or right, produced more high-arousal negative emotional content compared to balanced sources. This high-arousal negative content also increased the likelihood of reposting for biased sources as opposed to balanced ones (but this trend did not hold for other types of emotional content). In recent years, the virality of high-arousal negative content rose, especially in balanced news sources and posts about politics.

I would like to commend the authors for a methodologically robust work. The study is well thought out. The results are perhaps not too surprising and mostly consistent with the pre-existing literature but it is still important to do the quantitative documentation of such trends. The attention to detail and the robust methodology is visible in many places of the paper like for instance when the authors triangulate their initial choice of news media bias ratings (from Ad Fontes) with another provider (AllSides). The LDA topic modeling to characterize the news sources sections driving the results was another nice touch. The statistical analysis is solid and some of the somehow counterintuitive results (like the increased high arousal negative affective content in balanced news sources) are an important contribution to the existing literature. I barely have much criticism to utter and my recommendation is to accept this paper with only minor revisions.

I would also like to commend again the authors because as a reader of their manuscript they come across as very impartial in their work, as it should be! This is unfortunately not always the case when reading these sorts of paper, and it feels like some authors have an agenda or some sort of viewpoint bias that they try to accommodate the results to. This is not the case at all in the current paper. It feels exquisitely neutral in that regard. 

We deeply appreciate this encouraging feedback (particularly given that this is a new line of inquiry for us)! 

Only a bit of minor constructive criticism for the authors at the risk of being pedantic. I think the weakest link in the paper are the visualizations where there is definitely room for improvement. All the detailed reporting about the different linear models used for the analysis in the text is fine but sometimes a compelling visualization can convey the point across very efficiently.

Display more results graphically. 

We appreciate this opportunity to improve and augment the figures (and apologize for their originally suboptimal resolution). 

Specifically, we have now cleaned and re-rendered the existing figures, adding color to indicate affect type. Additionally, based on the feedback, we now have updated and integrated some of the figures from the Supplement into the manuscript (i.e., those depicting the validity of the bias ratings, a sentiment analysis flowchart, and a breakdown of the bias by time interaction). 

The comments in the discussion section about the spread of negative affect news content through the population is highly relevant and important. Perhaps the authors should mention at least a sort of counter hypothesis: that people perhaps could be developing some sort of resistance or immunity after perhaps too much exposure to negative sentiment in their news diets and could be tuning out. I'm not saying that this is happening, but perhaps the authors should concede this as one of many potential hypotheses.

Address potential alternative hypotheses. 

The notion that people habituate to high arousal negative content may have come from the figure showing that the difference in virality of high arousal negative content has narrowed over 10 years. This diminishing difference, however, is due to high arousal content becoming more viral for balanced sources rather than less viral for biased sources, increasing the overall virality of high arousal negative content over 10 years (which would be more consistent with sensitization than habituation). To clarify this increase, we now include a Figure from the Supplement (Figure 5) in the main manuscript which depicts this interaction. We also now clarify that both the overall production and virality of high arousal negative content increased over 10 years in the Discussion: 

“The prevalence and spread of high arousal negative content in biased sources was not limited to current developments or events, since it generally increased over ten years. Further, the virality of high arousal negative content did not decrease over time for biased news sources, but rather increased over time for balanced news sources.” (line 487)

Finally, the usage of SentiStrength is well reasoned. I was just wondering whether an LLM like gpt-3.5-turbo would have a high degree of interrater agreement with the SentiStrength algorithm annotations, thus providing further evidence for the validity of the findings. But again, this is a bit pedantic.

Consider the application of large language models to sentiment coding. 

We agree that these developments offer a promising set of methods for future inquiry (as further suggested by a new reference), and so now explicitly raise and recommend this research direction in the Discussion: 

“These findings coincide with a rise in scientific research on affective phenomena due to advances in measurement (including sentiment analysis), which allow researchers to test qualitative predictions with quantitative tools [35]. Recently, researchers have developed even more precise and granular tools for sentiment analysis (e.g., Large Language Models), which might be leveraged to test the specificity as well as the generalizability of these findings [36].” (line 552)

To conclude, well done to the authors for a solid piece of aca

---

## [Editor Report · Decision Letter 1]

17 Sep 2024

News source bias and sentiment on social media

PONE-D-24-21176R1

Dear Dr. Knutson,

We’re pleased to inform you that your manuscript has been judged scientifically suitable for publication and will be formally accepted for publication once it meets all outstanding technical requirements.

Kind regards,

Silvio Eduardo Alvarez Candido

Academic Editor

PLOS ONE

---

## [Editor Report · Acceptance letter]

20 Sep 2024

PONE-D-24-21176R1 

PLOS ONE

Dear Dr. Knutson, 

I'm pleased to inform you that your manuscript has been deemed suitable for publication in PLOS ONE. Congratulations! Your manuscript is now being handed over to our production team.

Kind regards, 

on behalf of

Dr. Silvio Eduardo Alvarez Candido 

Academic Editor

PLOS ONE